# REDUCE, REUSE, REINTERPRET: AN END-TO-END PIPELINE FOR RECYCLING PARTICLE PHYSICS RESULTS

## A PREPRINT

Giordon Stark[*,1], Camila Aristimuno Ots[1,2], and Mike Hance[†,1]

[1]University of California, Santa Cruz, Santa Cruz Institute for Particle Physics, 1156 High Street, Santa Cruz, CA 95064
[2]University of Southern California, Liquid Propulsion Laboratory, 854 Downey Way, Los Angeles, CA 90089

February 2, 2024

## ABSTRACT

Searches for new physics at the Large Hadron Collider have constrained many models of physics beyond the Standard Model. Many searches also provide resources that allow them to be reinterpreted in the context of other models. We describe a reinterpretation pipeline that examines previously untested models of new physics using supplementary information from ATLAS Supersymmetry (SUSY) searches in a way that provides accurate constraints even for models that differ meaningfully from the benchmark models of the original analysis. The public analysis information, such as public analysis routines and serialized probability models, is combined with common event generation and simulation toolkits MADGRAPH, PYTHIA8, and DELPHES into workflows steered by TOML configuration files, and bundled into the mapyde Python package. The use of mapyde is demonstrated by constraining previously untested SUSY models with compressed sleptons and electroweakinos using ATLAS results.

## 1 Introduction

Direct searches for new phenomena at the Large Hadron Collider (LHC) have constrained many models of beyond-the-Standard-Model (BSM) physics. A typical LHC search focuses on an experimental signature chosen for its ability to discriminate between SM and BSM sources, and benchmarks the results of the search using one or more specific models. In some cases the chosen benchmarks are representative models that solve a particular problem in particle physics, while in other cases the benchmark models are simplified models [1] that represent a broader (but still limited) parameter space within a theoretical framework such as supersymmetry (SUSY). In either case, the experimental results as published by LHC collaborations are only strictly applicable to their benchmark models, leaving the vast majority of BSM parameter space unexplored. Leveraging the full power of LHC data in the search for new physics requires tools that facilitate the re-use of those experimental results to test models that were not considered in the original search. The challenge to experiments is to publish or provide enough information to enable such efforts, and the challenge to the rest of the community is to use that information to extend our understanding of what BSM theories are still viable. These challenges form the reinterpretation problem.

Existing toolkits solve the reinterpretation problem in various ways, as described in Section III of Ref. [2] and in Refs. [3, 4]. Some toolkits implement existing analysis workflows in independent software frameworks which are more simulation-based: CHECKMATE [5], MADANALYSIS5 [6–8], GAMBIT's COLLIDERBIT [9–12], RIVET [13, 14], and Contur [15] (which interprets RIVET outputs). Others match simplified versions of full models to experimental results using efficiency maps, relying more on experimental data: SMODELS [16] and RECAST-based approaches [17] such as in Refs. [18–22]. The CMS collaboration provides Simplified Likelihood [23] correlation/covariance matrices for some analyses, while the ATLAS collaboration has started to provide full probability models [24] in which additional data is

---

[*]gistark@ucsc.edu
[†]mhance@ucsc.edu

Figure 1: Overview of the `mapyde` toolchain and the role of each component.

encoded, such as correlations and background estimates. The RECAST framework [17] facilitates full production and processing of simulated events using the same software tools used by ATLAS in the physics results of interest. This has the advantage of high accuracy and precision, but requires significant computational resources to produce fully simulated and reconstructed samples.

Recent progress in releasing full public probability models has further expanded the possibilities for reinterpreting LHC analyses. It is no longer a technical challenge to assess the sensitivity of an LHC analysis with a public serialized probability model, including all uncertainties and correlations, to an arbitrary model of new physics. The only challenge that remains is to evaluate the acceptance and efficiency of the analysis to the new model. This can often be done using public event generation and detector simulation tools, informed by the experimental details of the reference analysis.

In this paper we present a new pipeline for calculating the constraints on new physics from existing analyses with public probability models. This pipeline is built using `mapyde` [25], a pure-Python package that chains public tools in HEP with a single configuration file. Additionally, `mapyde` provides a user-friendly interface to configure, run, and extend the toolchain with other tools as needed. The `mapyde` toolkit is described in Section 2, including the software employed, the configuration, and deployment in containerized environments. Two example uses of the `mapyde` toolkit are provided in Section 3, in which we reproduce existing ATLAS results, test a new simplified model of slepton-wino-bino production, and run a pMSSM-like scan of electroweak SUSY model parameters to test SUSY models with mixed wino-bino-higgsino states.

## 2   The `mapyde` toolkit

This paper describes the implementation of, and results obtained with, version v0.5.0 of `mapyde`. This package can be installed via `pip` [26] or `conda` [27], as shown in Listing 1. Listing 2 provides some examples of how the authors intend this software to be executed. The `mapyde` package provides support for using the following tools:

- MADGRAPH5_AMC@NLO (MADGRAPH) [28, 29] (event generation)
- PYTHIA8 [30] (parton shower, hadronization, decays)
- DELPHES [31–33] (detector simulation)
- SimpleAnalysis [34, 35] (analysis description)
- pyhf [36, 37] (probability model fitting)

and was named after the first three tools in a typical `mapyde` simulation pipeline: MADGRAPH, PYTHIA8, and DELPHES (MAPYDE). Additional tools, such as those listed in Section 1, can be supported in the analysis pipeline of this flexible framework with a little extra custom configuration, and they can be more natively supported in future versions of

mapyde upon request or by pull requests from external contributors. A flow chart illustrating the role of different tools in the mapyde pipeline is shown in Figure 1.

```
    python -m pip install mapyde            # via pypi
    pipx install mapyde                      # via pypi
    conda install -c conda-forge mapyde     # via conda-forge
    mamba install mapyde                     # via conda-forge
```
Listing 1: Snippet illustrating various ways to install mapyde.

A mapyde pipeline is constructed as a series of containerized transforms, with the mapyde package providing configuration and steering of the individual stages. The use of containers allows the mapyde package to remain lightweight and easy to install, while still enabling event generation/simulation and subsequent data analysis steps to run on any system that provides Docker or Singularity/Apptainer. Default containers are provided for MADGRAPH+PYTHIA8, for DELPHES, and for probability model fitting with pyhf. The ATLAS collaboration provides containers that implement the SimpleAnalysis routines for many SUSY searches, including those reinterpreted in Section 3. Containers for a variety of different MADGRAPH+PYTHIA8 releases are provided in the mapyde GitHub container registry [38]. Users can also provide their own containers for any stage of the analysis.

```
    $ mapyde config parse user.toml   # parse and print configuration to screen
    $ mapyde --prefix cards           # print the on-disk path for configuration cards
    $ mapyde run all user.toml        # run the default workflow for user.toml
    $ mapyde run madgraph user.toml   # run the madgraph step for user.toml
```
Listing 2: Some example commands for running mapyde v0.5.0 from the command-line. From top to bottom, (a) parse the entire configuration and print a JSON-serialized representation to the screen, (b) print the prefix for where to find configuration cards on the installed machine, (c) run the default workflow steps & tools provided by mapyde for the given configuration, and (d) run only MADGRAPH step of the workflow using the provided configuration.

The mapyde workflow is controlled by a user-generated dictionary that encodes the configuration for all stages of the analysis. Templates for this dictionary are provided, and custom templates can be used to provide a consistent reference for subsequent analysis runs that modify a small number of configuration parameters. The configuration dictionary offers direct access to selected parameters in the MADGRAPH run card, in addition to allowing the user to point towards custom process and model parameter cards. Analyses performed with mapyde can use either a command-line interface or take advantage of direct access to Python functions. The command-line interface is implemented with the click Python package [39], and it takes a TOML [40] configuration file as input, which is translated internally into the configuration dictionary. As shown in Figure 2, mapyde also provides a *Text User Interface* (TUI) using Textual [41] to support users less familiar with the command-line interface. The Python interface to mapyde can either parse a TOML input file or accept the configuration dictionary directly. An example TOML configuration file is provided in Appendix C.

In the containers provided by mapyde, MADGRAPH and PYTHIA8 are bundled together and are both launched through a MADGRAPH control file that is generated based on the mapyde configuration dictionary. The MADGRAPH outputs are stored as Les Houches Event (LHE) records [42], and are passed directly into PYTHIA8 for parton showering and hadronization. The PYTHIA8 outputs in HEPMC format are then passed into DELPHES, or directly into SimpleAnalysis when evaluating the acceptance of an analysis selection without detector simulation. Outputs from DELPHES in ROOT format are passed to an intermediate stage that either analyzes the DELPHES output or transforms it into a format able to be processed by the next step in the pipeline. In the results presented below, we use a script that converts the DELPHES output into a ROOT file that can serve as input to the SimpleAnalysis phase, called Delphes2SA.py. The output of SimpleAnalysis is then processed into json format in another custom script called SA2JSON.py. That script encodes a mapping from the branch names in the SimpleAnalysis output file to the signal region names in the public probability model provided with the published analysis. The result of that transform can then be used to patch the serialized probability model and perform hypothesis tests with the generated signal. In the cases presented below, we perform the hypothesis test with the muscan.py script, whose output is a json file containing both the 95% confidence level upper limits on the cross section (expressed as a ratio of the cross section to the model prediction, called the signal strength and denoted $\mu_{\text{sig}}$), as well as the full configuration dictionary used for the job. These outputs can then be used to determine if a given set of parameters used to generate events for the hypothesis

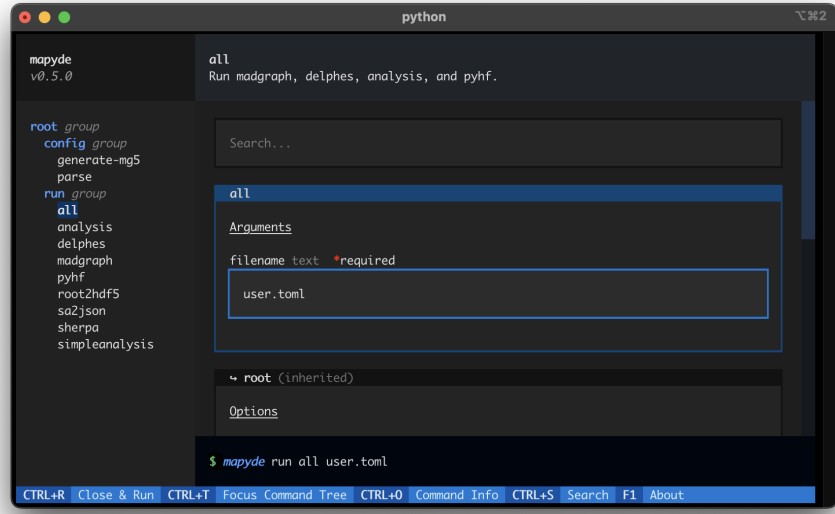

Figure 2: Screenshot of an iTerm2 terminal showing the Textual User Interface option of `mapyde` called from the command-line via `mapyde --tui`.

test are compatible with existing limits or not. All outputs from intermediate stages of the analysis chain are kept to facilitate re-running the pipeline from an arbitrary starting point.

## 3    Reinterpreting Compressed SUSY Searches from ATLAS

We demonstrate the utility of the `mapyde` analysis chain by reproducing and reinterpreting the results of ATLAS searches for supersymmetry. The searches described in Ref. [43] are optimized for SUSY models with "compressed" mass spectra, where the next-to-lightest SUSY partner (NLSP) and lightest SUSY partner (LSP) are separated by $\mathcal{O}(1-10)$ GeV in mass. The small mass splittings imply low-momentum (soft) Standard Model decay products, since most of the momentum of the NLSP is given to the LSP. The ATLAS searches considered here focus on final states including two low-momentum ("soft") charged leptons, substantial missing transverse momentum ($E_{\mathrm{T}}^{\mathrm{miss}}$) from the invisible LSP's (which in this case are the lightest neutralinos, $\widetilde{\chi}_1^0$), and one or more energetic jets that boost the SUSY system. We focus specifically on two searches from that paper: a search optimized for light sleptons (SUSY partners of the SM leptons), and a search optimized for light electroweakinos (SUSY partners of the SM electroweak bosons).

### 3.1    Implementation

We use `mapyde` to generate, shower, and simulate SUSY events, to analyze them using `SimpleAnalysis`, and to interpret the results using `pyhf`. The implementation corresponds to `mapyde` version 0.5.0, with configuration cards and scripts provided in a public `GitHub` repository [44]. The configuration uses MADGRAPH version 2.9.3, PYTHIA8 version 8.306, DELPHES version 3.5.0, and `SimpleAnalysis` version 1.1.0. The `SimpleAnalysis` code runs on dedicated containers provided by ATLAS [45], where we use the "EwkCompressed2018" selection corresponding to the analyses from Ref. [43]. We use `pyhf` version 0.7.2 to patch the public probability models provided in the HEPData [46] repository [47] for Ref. [43] with the signal yields from `mapyde`, and to compute upper limits on $\mu_{\mathrm{sig}}$. When reproducing the results for the same benchmark models from Ref. [43] the `pyhf` output is compared with limit contours provided in HEPData.

The signal samples produced in our `mapyde` workflow differ from those used in Ref. [43] in three significant ways. First, events in the ATLAS samples contained up to two jets in the matrix element in addition to a pair of SUSY particles, with different jet multiplicity processes merged in the parton shower using the CKKW-L algorithm [48]. In our samples, we produce only one-jet events in MADGRAPH, and allow PYTHIA8 to model the contributions from additional QCD emissions. Second, the ATLAS samples use MADSPIN [49] to perform the three-body decays of the electroweakinos to SM leptons and a $\widetilde{\chi}_1^0$, while we perform decays using PYTHIA8. (The `mapyde` pipeline does support the use of

MadSpin, but it is not used here.) Third, and most importantly, the ATLAS samples use ATLAS simulation and reconstruction software to transform the Pythia8 output into ROOT files containing physics objects, while we use Delphes.

The impact of the first two differences is evaluated by comparing the acceptance of the event selection applied to events at particle level. ATLAS provides the acceptance of the event selection by processing particle-level events with SimpleAnalysis as part of the public event record of the search. We perform a similar calculation by passing the HEPMC event record from Pythia8 directly into SimpleAnalysis using mapyde. The resulting signal region yields are then compared to the product of the ATLAS acceptance, cross section, branching ratio [50–55], and integrated luminosity. On average we find that the mapyde and ATLAS acceptances are very similar, except for very compressed points where on average the ATLAS acceptance is approximately 10% higher.

The final difference between the ATLAS simulation framework and mapyde is evaluated using signal yields and model constraints after detector simulation and tuned by modifying the efficiencies in the Delphes configuration card. We focus in particular on the treatment of electrons and muons in Delphes, since these also required special handling in the ATLAS analysis. Further details of the lepton efficiency tuning in Delphes are described below.

## 3.2    Compressed Sleptons

We first reproduce the results of the ATLAS search for compressed sleptons. The ATLAS search is optimized using a slepton-bino model, in which the slepton NLSP decays to a bino-like LSP and a charged SM lepton. We generate events with pairs of charged sleptons (including scalar superpartners of both left- and right-handed SM leptons) and compare the leading-order (LO) cross sections calculated by MadGraph in an inclusive sample (without additional emissions) with the next-to-leading-order (NLO) cross sections reported by ATLAS, which do not include electroweak corrections. We find a mass-independent NLO:LO $k$-factor of 1.18, which is used to scale the one-jet MadGraph samples produced with mapyde.

After the acceptance corrections described above, the lepton efficiencies in Delphes are tuned to reproduce the ATLAS signal yields as documented in Listings 4 and 5. The efficiencies of electrons and muons after object selection are provided as part of the public record for Ref. [43] and are the starting point for the modified Delphes configuration used to reproduce the ATLAS results. We find that setting all electron and muon efficiencies in Delphes to the upper range of values reported in Fig. 3 of [43] is sufficient to reproduce the results of the ATLAS slepton and electroweakino searches to adequate precision, as shown in Figures 3 and 8. The only exception is the lowest-$p_{\mathrm{T}}$ bin for both electrons and muons, where the mapyde efficiencies needed to reproduce the ATLAS limits are roughly 15% higher than the values reported by ATLAS. Figure 3 also shows the model constraints for the slepton-bino search when using the default Delphes configuration card, which does a reasonable job of describing the high-splitting regions, but fails to describe the most compressed mass points, which are most sensitive to the low-$p_{\mathrm{T}}$ lepton efficiencies provided by ATLAS.

With the framework tuned to the ATLAS response for compressed SUSY events, we now use mapyde to assess the sensitivity of the ATLAS search to a new model. We consider a "slepton-wino-bino" simplified model, where a light slepton decays to a wino and a SM charged lepton, as illustrated in Figure 4. We set the slepton branching ratio $\tilde{\ell} \to \ell \tilde{\chi}_2^0$ to 100%, with the $\tilde{\chi}_2^0$ decaying to a $\tilde{\chi}_1^0$ and to two fermions through an off-shell $Z$ boson.[3] In such events, the $p_{\mathrm{T}}$ values of the SM leptons will be largely determined by the $\tilde{\ell}$–$\tilde{\chi}_2^0$ mass gap, in contrast to the slepton-bino model where the lepton $p_{\mathrm{T}}$ is driven by the $\tilde{\ell}$–$\tilde{\chi}_1^0$ mass gap. In the limit of vanishing $\tilde{\chi}_2^0$–$\tilde{\chi}_1^0$ mass differences, the slepton-wino-bino model described here is phenomenologically identical to the slepton-bino model.

We parameterize the model in terms of the slepton mass, the $\tilde{\ell}$-$\tilde{\chi}_1^0$ mass gap, and the $\tilde{\chi}_2^0$-$\tilde{\chi}_1^0$ mass gap, to allow us to display the model constraints in the same plane as the slepton-bino results from ATLAS. The resulting constraints (both expected and observed) are shown in Figures 6 and 7 for a range of $\tilde{\chi}_2^0$-$\tilde{\chi}_1^0$ splittings. At small $\Delta m(\tilde{\chi}_2^0, \tilde{\chi}_1^0)$ the results resemble those of the ATLAS slepton-bino model, since the $\tilde{\chi}_2^0$ is nearly degenerate with the $\tilde{\chi}_1^0$ leading to virtually no kinematic difference arising from the additional soft decay products. By contrast, for large $\tilde{\chi}_2^0$-$\tilde{\chi}_1^0$ splittings (dark blue contour lines), the requirement that the slepton must decay through a $\tilde{\chi}_2^0$ results in stronger constraints at large $\tilde{\ell}$-$\tilde{\chi}_1^0$

---

[3]We ignore decays of the slepton through a chargino, despite them likely being present in "reaslistic" slepton-wino-bino scenarios, for two reasons. First, the charged leptons produced in such a decay chain are hurt by low branching fractions and softer kinematics, while charged leptons are directly produced in the slepton$\to \tilde{\chi}_2^0$ decay and acquire a larger fraction of the slepton momentum and are easier to reconstruct. Taken together this means that the contributions to the signal region yields from slepton$\to \tilde{\chi}_1^\pm$ decays are very small; we found that they can be ignored entirely if the slepton$\to \tilde{\chi}_2^0$ branching fraction is non-negligible. Second, with 100% slepton$\to \tilde{\chi}_2^0$ branching ratios, the slepton-wino-bino model naturally converges to the simpler slepton-bino model when the $\tilde{\chi}_2^0$ and $\tilde{\chi}_1^0$ are very close in mass, allowing us to build intuition about how small variations in SUSY models affect the corresponding constraints.

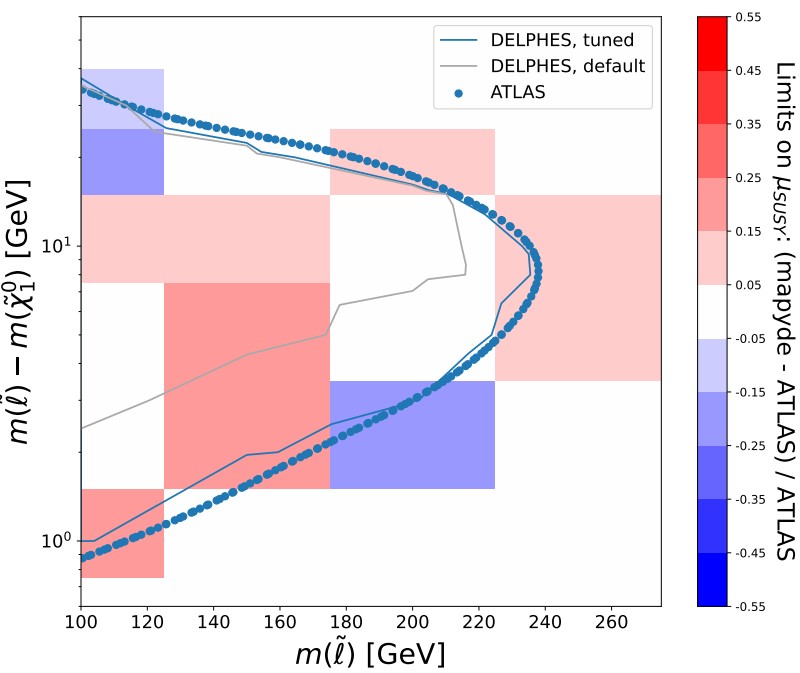

Figure 3: Constraints on the slepton-bino model from: the ATLAS paper [43] (blue dots); `mapyde` before tuning the DELPHES lepton efficiencies (gray line); and `mapyde` after tuning (blue line). The color map shows the relative difference in the limits on the SUSY signal strength, $\mu_{\text{SUSY}}$, between `mapyde` and ATLAS results.

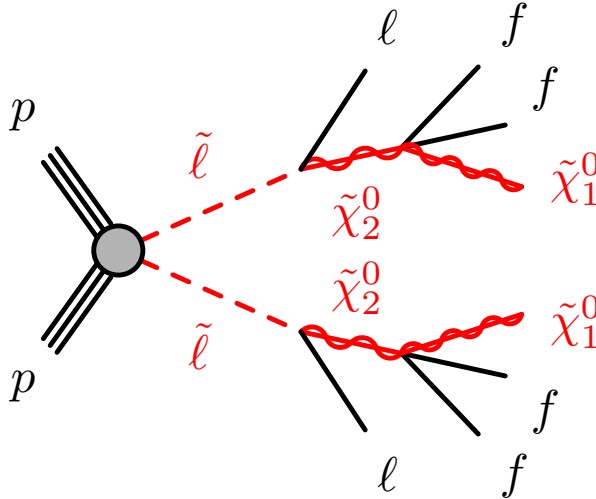

Figure 4: Feynman diagram of slepton pair production, with slepton ($\tilde{\ell}$) decays to wino-like NLSP's ($\tilde{\chi}_2^0$), which then decay to bino-like LSP's ($\tilde{\chi}_1^0$).

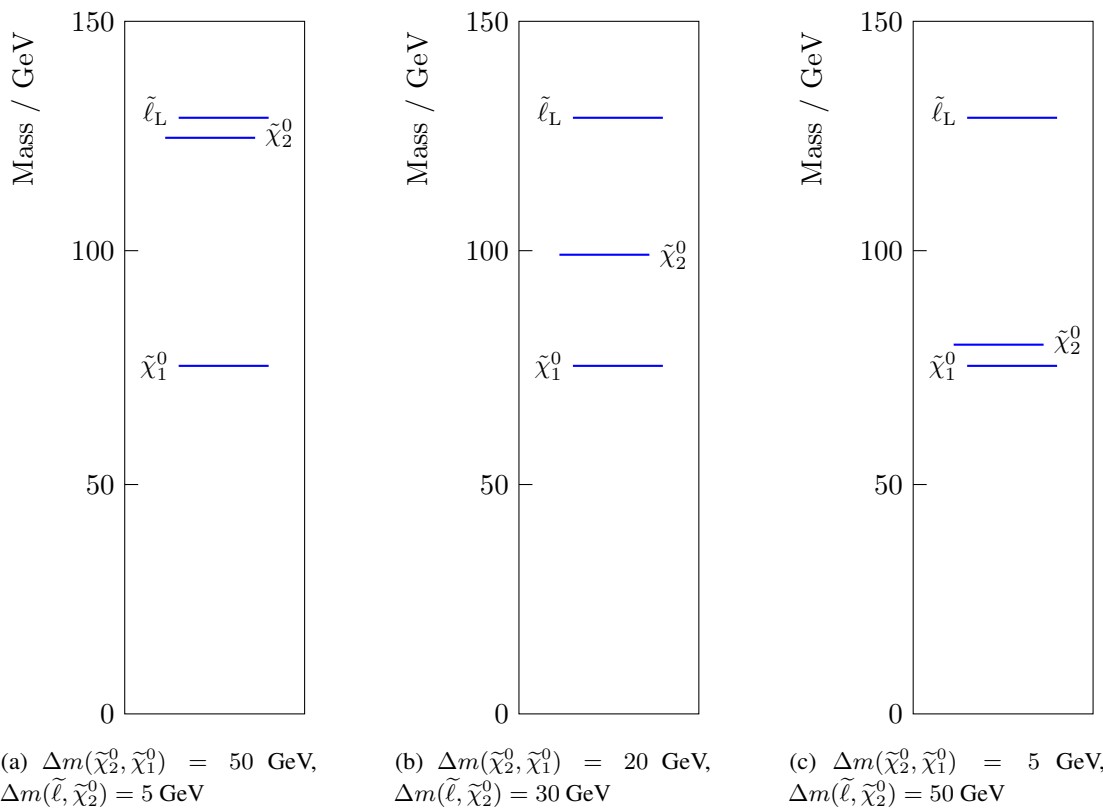

(a) $\Delta m(\widetilde{\chi}_2^0, \widetilde{\chi}_1^0) = 50$ GeV, $\Delta m(\widetilde{\ell}, \widetilde{\chi}_2^0) = 5$ GeV

(b) $\Delta m(\widetilde{\chi}_2^0, \widetilde{\chi}_1^0) = 20$ GeV, $\Delta m(\widetilde{\ell}, \widetilde{\chi}_2^0) = 30$ GeV

(c) $\Delta m(\widetilde{\chi}_2^0, \widetilde{\chi}_1^0) = 5$ GeV, $\Delta m(\widetilde{\ell}, \widetilde{\chi}_2^0) = 50$ GeV

Figure 5: Mass hierarchies for large $\Delta m(\widetilde{\ell}, \widetilde{\chi}_1^0)$ splittings for different splittings of $\Delta m(\widetilde{\chi}_2^0, \widetilde{\chi}_1^0)$ being (a) large, (b) intermediate, and (c) small. In all figures, $m(\widetilde{\ell}) = 130$ GeV.

differences. For each model considered, the contour is prevented from covering arbitrarily small values of $\Delta m(\widetilde{\ell}, \widetilde{\chi}_2^0)$ by the fact that such models produce softer SM leptons and lower lepton efficiencies. Similarly, the contour is bounded from above by the reduced cross-section due to larger slepton masses, and by ATLAS analysis selections, which were optimized for compressed decays, and which are sensitive to the momenta of the fermions from the $\widetilde{\chi}_2^0$ decay. For models with large $\Delta m(\widetilde{\ell}, \widetilde{\chi}_1^0)$, the second-lightest neutralino $\widetilde{\chi}_2^0$ has more allowed phase-space illustrated in Figure 5.

This class of models, while being less "simplified" than the slepton-bino model traditionally studied by LHC experiments, includes models that nominally escape constraint by both existing slepton-bino searches as well as direct searches for electroweakinos. In particular, the wino-bino constraints reported in Ref. [43] constrain electroweakinos with $\widetilde{\chi}_2^0$-$\widetilde{\chi}_1^0$ splittings as low as 1 GeV only for electroweakino masses near 100 GeV. In the slepton-wino-bino results shown in Fig. 6 we find that the ATLAS search excludes models with 1 GeV $\widetilde{\chi}_2^0$-$\widetilde{\chi}_1^0$ splittings up to slepton masses of well over 200 GeV, which implies limits on $\widetilde{\chi}_1^0$ masses also exceed 200 GeV for small slepton-$\widetilde{\chi}_1^0$ splittings.

### 3.3    Compressed Electroweakinos

The ATLAS search for compressed electroweakinos considered two different simplified models: one with "pure" Higgsino-like $\widetilde{\chi}_2^0$, $\widetilde{\chi}_1^\pm$, and $\widetilde{\chi}_1^0$ states, and another with wino-like $\widetilde{\chi}_2^0/\widetilde{\chi}_1^\pm$ and bino-like $\widetilde{\chi}_1^0$. We focus on the Higgsino model to validate the mapyde output. Following a procedure similar to the validation of the ATLAS slepton-bino results described above, we generate a grid of Higgsino model points to reproduce the ATLAS results. The mapyde electroweakino samples use the same $k$-factor and lepton efficiencies as the slepton search. The branching ratio of the Higgsino-like $\widetilde{\chi}_2^0$ to leptons is taken from LHC SUSY Cross Section Working Group [56]. The comparison between the mapyde and ATLAS results is shown in Figure 8, where the mapyde exclusion contour follows the corresponding ATLAS contour.

We next use mapyde to assess the ATLAS sensitivity to MSSM SUSY models in which the bino, wino, and Higgsino mass terms ($M_1$, $M_2$, and $\mu$, respectively) are all relatively low, leading to "wino-bino-Higgsino" models that have potentially rich phenomenologies. Some example models are illustrated in Figure 9. In these models, the $W$-boson

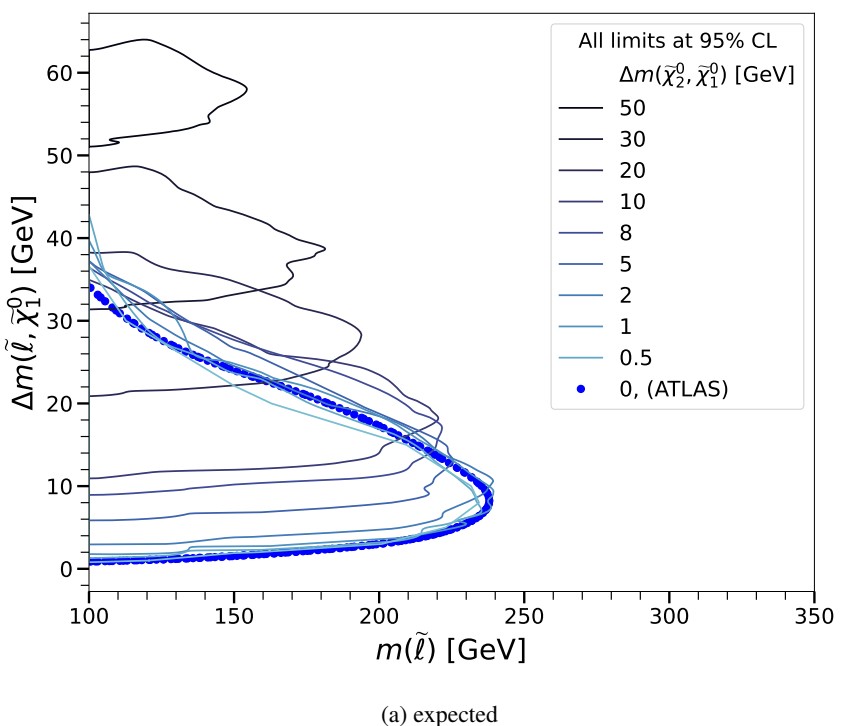

(a) expected

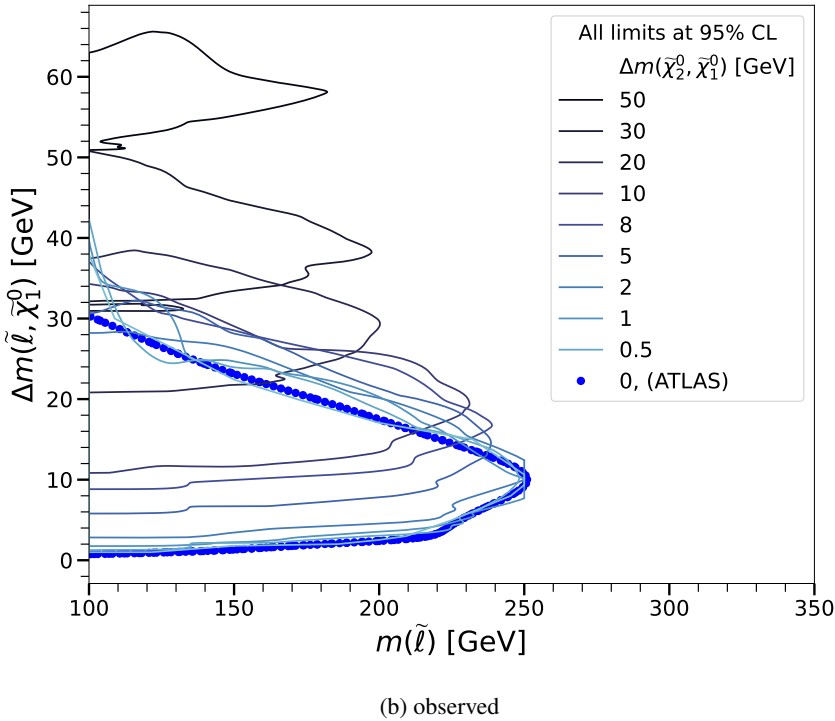

(b) observed

Figure 6: Expected (a) and observed (b) constraints on the slepton-wino-bino model. The model is parameterized by the slepton mass, the slepton-$\widetilde{\chi}_1^0$ mass splitting, and the $\widetilde{\chi}_2^0$-$\widetilde{\chi}_1^0$ splitting, and compared against the slepton-bino results from Ref. [43].

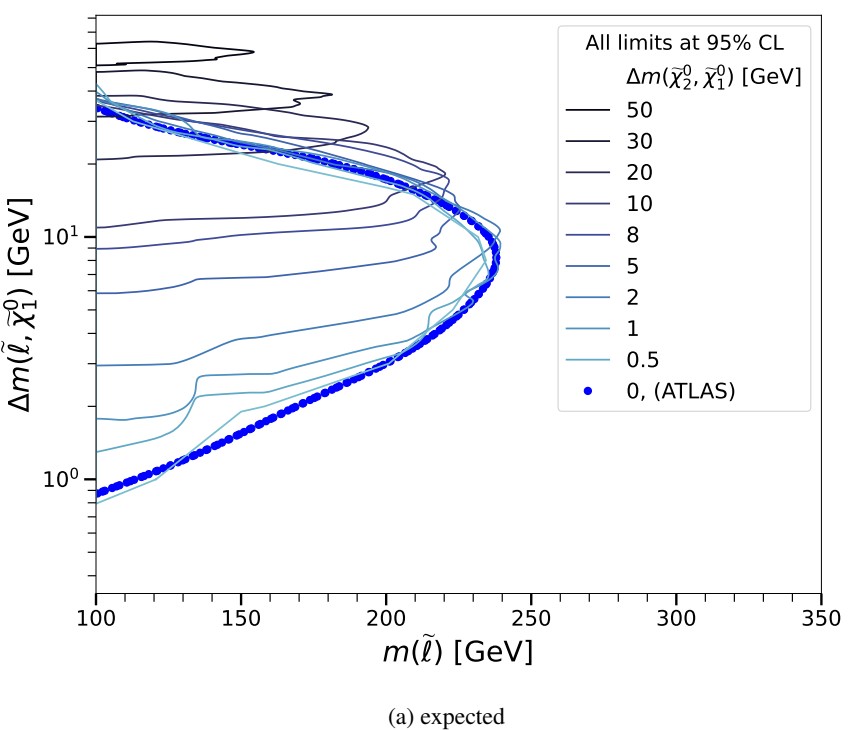

(a) expected

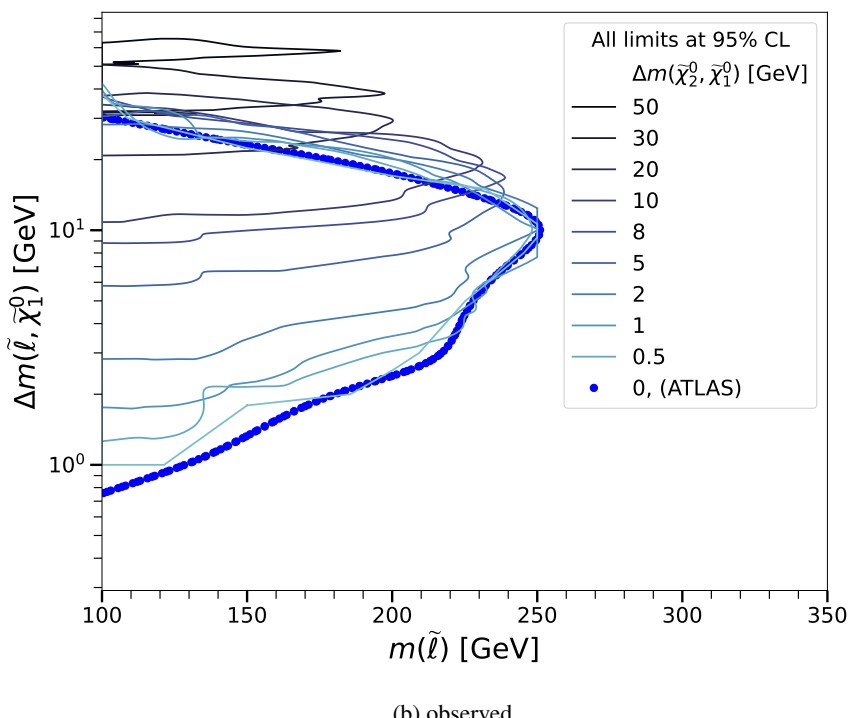

(b) observed

Figure 7: Expected (a) and observed (b) constraints on the slepton-wino-bino model, shown in logarithmic scale. The model is parameterized by the slepton mass, the slepton-$\widetilde{\chi}_1^0$ mass splitting, and the $\widetilde{\chi}_2^0$-$\widetilde{\chi}_1^0$ splitting, and compared against the slepton-bino results from Ref. [43].

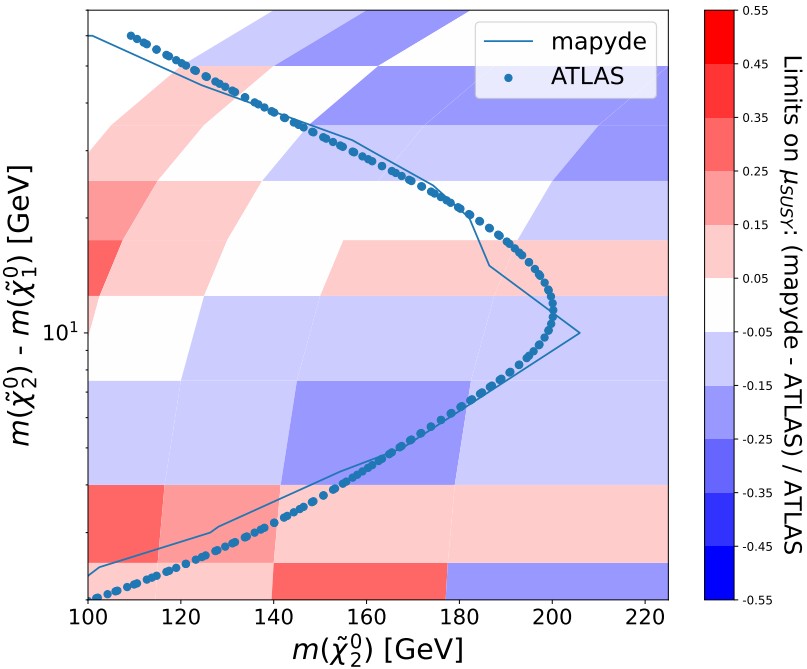

Figure 8: Constraints on the "pure Higgsino" simplified model from ATLAS (blue dots) and mapyde (blue line). The color map shows the relative difference in the limits on the SUSY signal strength, $\mu_{\mathrm{SUSY}}$, between mapyde and ATLAS results.

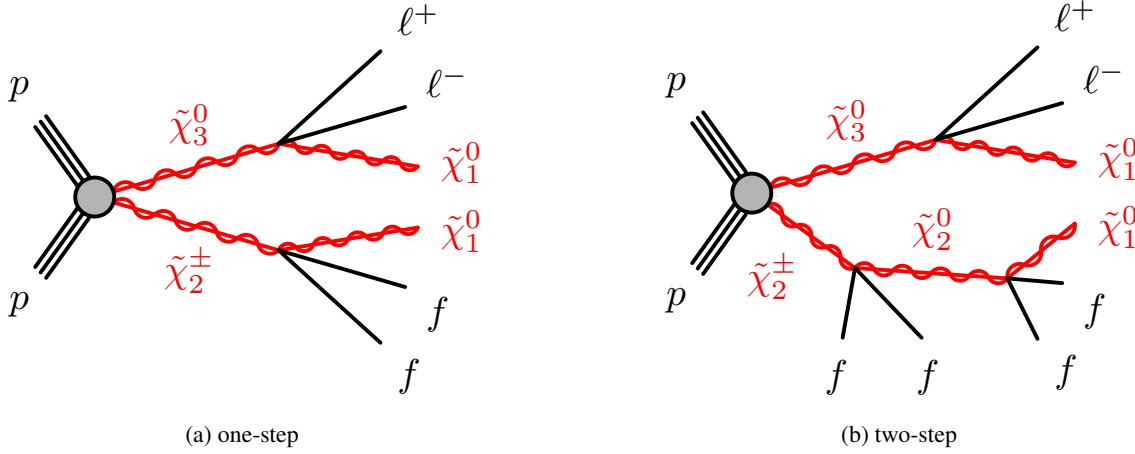

(a) one-step

(b) two-step

Figure 9: Example feynman diagrams of possible electroweakino decays with (a) $\widetilde{\chi}_2^\pm \rightarrow \widetilde{\chi}_1^0$ and (b) $\widetilde{\chi}_2^\pm \rightarrow \widetilde{\chi}_2^0$.

is treated as off-shell for the "compressed" phase-space under study in this paper. We use a pMSSM scanning tool (EASYSCAN_HEP [57, 58]) to generate particle spectra for models with $M_1$, $M_2$, and $\mu$ ranging from $-500$ GeV to 500 GeV, sampled with flat priors. Particle masses are calculated using SPHENO [59, 60] and stored in the SUSY Les Houches Accord (SLHA) format [61]. Additional tools are run as described in Appendix B and are used to calculate values for the selection criteria shown in Listing 3. Since our goal is to investigate models with compressed electroweakino mass spectra accessible to ATLAS Run-2 searches, we select models that satisfy $m(\widetilde{\chi}_1^0) > 100$ GeV, $m(\widetilde{\chi}_3^0) < 300$ GeV, and $(m(\widetilde{\chi}_3^0) - m(\widetilde{\chi}_1^0)) < 50$ GeV for further study. We further require that any selected models have valid output from the spectrum generator SPHENO [59, 60], have a valid Higgs mass as computed by FEYNHIGGS [62–69], and to satisfy dark matter constraints ($\Omega h^2 < 0.12$) implemented in MICROMEGAS [70], flavor physics constraints implemented in SUPERISO [71], and optionally muon $g - 2$ constraints implemented in GM2CALC [72, 73]. The SLHA records for the selected 81 models are used as inputs for event generation with `mapyde`. Branching ratios for electroweakino decays are taken directly from the SPHENO output.

```
1    (SP_m_h!=-1) & (SPfh_m_h!=-1) # spheno, feynhiggs: ok
2    & (SI_BR_Bs_to_mumu!=-1) & (GM2_gmuon!=-1) # superiso, gm2calc: ok
3    & (MO_Omega!=-1) & (MO_Omega < 0.12) # micromegas: ok, DM relic density
4    & (SP_m_chi_10>100) & (abs(SP_m_chi_30)<300) # N1 > 100 GeV, N3 < 300 GeV
5    & ((abs(SP_m_chi_30)-abs(SP_m_chi_10))<50) # m(N3, N1) < 50 GeV
```

Listing 3: The mask used to define the selection of models for assessing the ATLAS sensitivity to Higgsino-Win We next use `mapyde` to assess the ATLAS sensitivity to MSSM SUSY models in which the bino, wino, and Higgsino o models.

The ATLAS constraints on the wino-bino-higgsino model scan are parameterized in terms of the mass of the $\widetilde{\chi}_2^0$ and the mass difference $\Delta m = m(\widetilde{\chi}_2^0) - m(\widetilde{\chi}_1^0)$, to facilitate comparisons with the pure-Higgsino constraints from ATLAS. The models under study are binned in a coarse grid in the $\Delta m$ vs $m(\widetilde{\chi}_2^0)$ plane, and the fraction of excluded models is calculated for each bin. The expected and observed constraints on these models are shown in Figure 10, with the ATLAS Higgsino results overlaid for comparison. In general the selected pMSSM points are more constrained than those of a pure-Higgsino model at larger mass splittings, likely due to the presence of the additional $\widetilde{\chi}_3^0$ and its own decays to final states similar to that of the $\widetilde{\chi}_2^0$.

# 4    Conclusion

The combination of rich physics results from the LHC with public supplementary material, including analysis routines and probability models, has opened the door to new ways of quantifying constraints on unexplored models of BSM physics with already-published results. We demonstrated one such pipeline, implemented in the `mapyde` Python package, that facilitates the re-use of LHC analyses. We illustrated the utility of a user-friendly Python package, `mapyde`, by probing previously-untested models of supersymmetry, including a simplified model of sleptons that undergo cascade decays to wino-like and bino-like electroweakinos, and a parameter scan of highly-mixed higgsino, wino, and bino states with a rich set of possible decay chains. In both cases we find that existing searches are able to constrain non-trivial regions of parameter space that would have been difficult or impossible to predict by simple extrapolations of existing results.

# 5    Acknowledgments

We thank Sam English, Bryn Lonsbrough, Len Morales Zaragoza, Zach Dethloff, and Ryota Johnson for suffering with early versions of scripts that would later develop into `mapyde`. We thank Zach Marshall, Jeff Shahinian, and Jason Nielsen for reading early drafts of this work and providing useful comments. Hance and Stark are supported by the Department of Energy Office of Science grant DE-SC0010107.

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

# A    Lepton Efficiencies in DELPHES

```
module Efficiency ElectronEfficiency {
  set InputArray ElectronFilter/electrons
  set OutputArray electrons

  # set EfficiencyFormula {efficiency formula as a function of eta and pt}
    set EfficiencyFormula {                    (pt <  4.5)               * (0.00) +
                          (abs(eta) <= 2.5) * (pt < 5.0) * (pt >= 4.5) * (0.30) +
                          (abs(eta) <= 2.5) * (pt < 6.0) * (pt >= 5.0) * (0.45) +
                          (abs(eta) <= 2.5) * (pt < 8.0) * (pt >= 6.0) * (0.52) +
                          (abs(eta) <= 2.5) * (pt < 10)  * (pt >= 8.0) * (0.65) +
                          (abs(eta) <= 2.5) * (pt < 20)  * (pt >= 10)  * (0.68) +
                          (abs(eta) <= 2.5) * (pt < 30)  * (pt >= 20)  * (0.70) +
                          (abs(eta) <= 2.5) * (pt < 50)  * (pt >= 30)  * (0.75) +
                          (abs(eta) <= 2.5) * (pt >= 50)               * (0.87) +
                      (abs(eta) > 2.5)                                 * (0.00)}
}
```

Listing 4: A snippet from the `mapyde`-provided DELPHES configuration for electron efficiencies. The portion that is commented out is what comes from the default ATLAS configuration. These numbers come from Ref. [43].

```
module Efficiency MuonEfficiency {
  set InputArray MuonMomentumSmearing/muons
  set OutputArray muons

  # set EfficiencyFormula {efficiency as a function of eta and pt}

  set EfficiencyFormula {                    (pt <  3.0)               * (0.00) +
                          (abs(eta) <= 2.7) * (pt < 3.5) * (pt >= 3.0) * (0.65) +
                          (abs(eta) <= 2.7) * (pt < 4.5) * (pt >= 3.5) * (0.72) +
                          (abs(eta) <= 2.7) * (pt < 7.0) * (pt >= 4.5) * (0.75) +
                          (abs(eta) <= 2.7) * (pt < 10)  * (pt >= 7.0) * (0.78) +
                          (abs(eta) <= 2.7) * (pt < 15)  * (pt >= 10)  * (0.80) +
                          (abs(eta) <= 2.7) * (pt < 20)  * (pt >= 15)  * (0.82) +
                          (abs(eta) <= 2.7) * (pt < 30)  * (pt >= 20)  * (0.85) +
                          (abs(eta) <= 2.7) * (pt < 50)  * (pt >= 30)  * (0.90) +
                          (abs(eta) <= 2.7) * (pt >= 50)               * (0.93) +
                      (abs(eta) > 2.7)                                 * (0.00)}
}
```

Listing 5: A snippet from the `mapyde`-provided DELPHES configuration for muon efficiencies. The portion that is commented out is what comes from the default ATLAS configuration. These numbers come from Ref. [43].

## B    EasyScan_HEP Configuration

We used an EASYSCAN_HEPini configuration file that defined the scan ranges in Listing 6 as well as additional programs to run, described in the text above. The specific versions used are:

- EASYSCAN_HEP (v1.0.0): pMSSM scanning and program control [57, 58]

- SPHENO (v4.0.4): spectrum generator [59, 60]

- FEYNHIGGS (v2.16.0): Higgs mass calculation [62–69]

- MICROMEGAS (v5.2.1): Dark Matter calculations (e.g., relic density) [70]

- SUPERISO (v4.0): Flavor Physics observables [71]

- GM2CALC (v2.0.0): $g - 2$ calculation [72, 73]

```
[scan]
Scan method:      random
#                 ID      Prior   Min     MAX
Input parametes:  tanb,   Flat,   1,      60
                  M_1,    Flat,   -500,   500
                  M_2,    Flat,   -500,   500
                  M_3,    Flat,   2000,   2000
                  AT,     Flat,   2000,   2000
                  Ab,     Flat,   2000,   2000
                  Atau,   Flat,   2000,   2000
                  MU,     Flat,   -500,   500
                  mA,     Flat,   2000,   2000
                  meL,    Flat,   2000,   2000
                  mtauL,  Flat,   2000,   2000
                  meR,    Flat,   2000,   2000
                  mtauR,  Flat,   2000,   2000
                  mqL1,   Flat,   2000,   2000
                  mqL3,   Flat,   2000,   2000
                  muR,    Flat,   2000,   2000
                  mtR,    Flat,   2000,   2000
                  mdR,    Flat,   2000,   2000
                  mbR,    Flat,   2000,   2000
```

Listing 6: A portion of the `easyscan.ini` configuration defining the random sampling for the electroweakinos scan.

## C    Input TOML Configuration for a Slepton Sample

The configuration dictionary used within `mapyde` can be created as a Python dictionary and passed to `mapyde` through the Python interface, or can be generated from a TOML configuration file. We provide an example TOML configuration file below. The code in this section represents a single file, but is described in blocks for easier interpretation.

### C.1    The `base` block

The `base` block provides `mapyde` with paths for inputs and outputs of the analysis pipeline. Users can take advantage of pre-generated configuration cards PYTHIA8 and DELPHES, but will usually at least need to define their own process cards for generating events with MADGRAPH.

```
[base]
path = "{{PWD}}"
output = "output"
logs = "logs"
data_path = "{{MAPYDE_DATA}}"
cards_path = "{{MAPYDE_CARDS}}"
scripts_path = "{{MAPYDE_SCRIPTS}}"
process_path = "{{MAPYDE_CARDS}}/process/"
param_path = "{{MAPYDE_CARDS}}/param/"
run_path = "{{MAPYDE_CARDS}}/run/"
pythia_path = "{{MAPYDE_CARDS}}/pythia/"
delphes_path = "{{MAPYDE_CARDS}}/delphes/"
madspin_path = "{{MAPYDE_CARDS}}/madspin/"
likelihoods_path = "{{MAPYDE_LIKELIHOODS}}"
```

Listing 7: The `base` block of an example TOML configuration file for generating slepton events.

Note that in Listing 7, the `template` is left undefined, which means that the base template shipped with `mapyde` will be loaded in. This behavior can be disabled by setting `template = false`. The (nested) template inheriting behavior for TOML configuration parsing enables defining grid scans more cleanly with less duplicated code.

## C.2 The `madgraph` block

The `madgraph` block defines some top-level options for running madgraph, such as:

- whether or not to skip the MADGRAPH job, sometimes useful when re-running an analysis chain and re-using LHE inputs from a previous job

- the name of the `param` card to use, in this example making use of a TOML feature that allows references to other TOML parameters

- the number of jobs to run in parallel when generating MADGRAPH events and when processing those events in PYTHIA8

- the name of the container that provides the version of MADGRAPH to use, in this case version 2.9.3 (taken from the GitHub container registry)

These options, and options from subsequent MADGRAPH-related blocks, are processed by `mapyde` to produce a MADGRAPH run script. The `madgraph.generator` block only specifies the name of that run script.

```
[madgraph]
skip = false
params = "SleptonBino"
paramcard = "{{madgraph['params']}}.slha"
cores = 1
batch = false
version = "madgraph:2.9.3"

[madgraph.generator]
output = "run.mg5"
```

Listing 8: The `madgraph` block of an example TOML configuration file for generating slepton events.

## C.3 The `madgraph.masses` block

The `madgraph.masses` block enables the use of user-defined substitutions in the SLHA param card.

```
[madgraph.masses]
MSLEP = 250
MN1 = 240
```

Listing 9: The `madgraph.masses` block of an example TOML configuration file for generating slepton events.

In the example above, the slepton masses in the SLHA file have been replaced with the string 'MSLEP', which mapyde then substitutes with the value of 250 GeV at run-time. This allows a single param-card template to be used for jobs that want to keep most parameters fixed, but vary one or more parameters as part of a parameter scan.

## C.4    The `madgraph.run` blocks

The `madgraph.run` blocks exposes the MADGRAPH run card to the mapyde configuration dictionary.    The `madgraph.run` block defines the name of the MADGRAPH run card, which will be accessed via the `run_path` defined in the `base` block as well as some commonly-changed run parameters: `nevents`, `iseed`, and `ecms`. The `madgraph.run.options` block defines modifications to be made to the MADGRAPH run card. In the example below, default values for `mmjj`, `ptj`, and `ptj1min` are overwritten by the values provided in the TOML configuration file.

```
[madgraph.run]
card = "default_LO.dat"
ecms = 13000
nevents = 50000
seed = 0

[madgraph.run.options]
mmjj = 500
ptj = 20
ptj1min = 50
```

Listing 10: The `madgraph.run` blocks of an example TOML configuration file for generating slepton events.

## C.5    The `madgraph.proc` block

The `madgraph.proc` block defines the name of the process card that contains the hard process information for MADGRAPH. Listing 11 shows an example using an existing process card, while  Listing 12 creates a new one from the specified `contents`.

```
[madgraph.proc]
name = "isrslep"
card = "{{madgraph['proc']['name']}}"
contents = false
```

Listing 11: The `madgraph.proc` block of an example full TOML configuration file for generating slepton events.

```
[madgraph.proc]
name = "isrslep"
card = false
contents = """\
set ...
define susystrong ...
define ...
generate p p > chsleptons chsleptons j / susystrong @1
output -f
"""
```

Listing 12: The `madgraph.proc` block demonstrating how to create a process card on-the-fly using `contents` to generate slepton events.

## C.6   The `madspin` block

The `madspin` block defines the name of the MADSPIN card, in jobs where MADSPIN is run. The running of MADSPIN can be skipped if it is not needed.

```
[madspin]
skip = true
card = ''
```

Listing 13: The `madgraph.run` block of an example TOML configuration file for generating slepton events.

## C.7   The `pythia` block

The `pythia` block holds all configuration information for PYTHIA8. It defines the name of the PYTHIA8 configuration file, controls whether PYTHIA8 is run or not, and allows the passing of some additional options that will be appended to the PYTHIA8 configuration card.

```
[pythia]
skip = false
card = "pythia8_card.dat"
additional_opts = ""
```

Listing 14: The `pythia` block of an example TOML configuration file for generating slepton events.

## C.8   The `delphes` block

The `delphes` block holds all configuration information for the DELPHES stage. It defines the name of the `tcl` configuration card, the name of the container that provides DELPHES(`version`), and the name of the output file. The input is assumed to be the `hepmc` output from a previous PYTHIA8 job.

```
[delphes]
skip = false
card = "delphes_card_ATLAS_lowptleptons_sleptons_notrackineffic.tcl"
version = "delphes"
output = "delphes/delphes.root"
```

Listing 15: The `delphes` block of an example TOML configuration file for generating slepton events.

## C.9   The `analysis` block

The `analysis` block defines any transform that comes after the DELPHES stage. In the example code below, the script `Delphes2SA.py` will transform the ROOT output from DELPHES into a ROOT file that can be processed by

SimpleAnalysis. This stage also allows the implementation of scale factors (`kfactor`) that are applied to the event weights from the DELPHES output. Scaling of the DELPHES outputs to a total integrated luminosity is also facilitated by the `lumi` flag. The cross section used to calculate event weights can be over-ridden with the `XSoverride` flag, which is multiplied by the `kfactor` to define the final cross section. Users can also provide their own scripts to run at this stage, which can choose to implement or ignore the options in this example configuration. Running multiple scripts can be accomplished in at least two different ways: by wrapping multiple scripts and having `mapyde` call the wrapper, or by calling `mapyde` multiple times with separate configuration files that differ only in the script name.

The code defined with the `script` flag is run within the same container as the DELPHES stage, to allow the use of DELPHES libraries for analyzing the output ROOT file.

```
[analysis]
script = "Delphes2SA.py"
XSoverride = -1
kfactor = 1.18
output = "analysis/Delphes2SA.root"
lumi = 139000
```

Listing 16: The `analysis` block of an example TOML configuration file for generating slepton events.

## C.10    The `simpleanalysis` block

The `simpleanalysis` block defines the inputs, outputs, and analysis code to be run in the `SimpleAnalysis` stage. The output filenames will be given the `name` of the `SimpleAnalysis` algorithm (here `EwkCompressed2018`, also see Listing 17), with the optional `outputtag` appended. This is particularly useful in cases where the input is specified as `hepmc`, indicating the output from PYTHIA8 should be analyzed, rather than the output from DELPHES. In such cases, specifying an `outputtag` like `_hepmc` can help to distinguish between those results and the results of running `SimpleAnalysis` on DELPHES output. Currently `mapyde` only supports the definition of a single `SimpleAnalysis` stage in the pipeline, so multiple `SimpleAnalysis` transforms, e.g., one for DELPHES inputs and one for HEPMC inputs, need to be performed using separate configurations.

```
#include "SimpleAnalysisFramework/AnalysisClass.h"

DefineAnalysis(EwkOneLeptonTwoBjets2018)
```

Listing 17: A snippet of SimpleAnalysisCodes/src/ANA-SUSY-2019-08.cxx [74] showing how the analysis name is defined.

```
[simpleanalysis]
name = "EwkCompressed2018"
input = ""
outputtag = ""
```

Listing 18: The `simpleanalysis` block of an example TOML configuration file for generating slepton events.

## C.11    The `sa2json` block

The `sa2json` block controls the translation of the `SimpleAnalysis` outputs in ROOT file format to a `json` patch file that can be used to update the probability model for statistical inference. This stage requires an understanding of the `SimpleAnalysis` output format, and how to relate that output to the names of signal regions defined in the public probability model from HEPData. An example is provided by the `SA2JSON.py` script in `mapyde`, but each serialized probability model will need a custom translation script. The `mapyde` developers welcome contributions of such scripts to the `mapyde` code base.

The `image` tag controls the name of the container to be used when running the transform, while the `input` and `output` options point to the `SimpleAnalysis` outputs and output `json` filename, respectively. The `options` entry collects

command-line options for the `SA2JSON.py` script. In this case, `-c` tells `SA2JSON.py` to perform a special selection for the compressed electroweak SUSY searches provided by the `EwkCompressed2018 SimpleAnalysis` selection.

```
[sa2json]
inputs = "{{simpleanalysis['name']}}{{simpleanalysis['outputtag']}}.root"
image = "pyplotting:latest"
output = "{{simpleanalysis['name']}}{{simpleanalysis['outputtag']}}_patch.json"
options = "-c"
```
Listing 19: The `sa2json` block of an example TOML configuration file for generating slepton events.

### C.12   The `pyhf` block

The `pyhf` block controls the options for running `pyhf` on an input probability model. The `image` flag controls the container to use for running `pyhf`. The serialized model, provided in `json` format, is specified with the `likelihood` flag. The `gpu-options` and `other-options` flags both pass command-line arguments to the script `muscan.py`, provided as part of the `mapyde` package, which performs a scan of the signal strength, $\mu_{\mathrm{sig}}$. If the container includes support for a GPU (such as the CUDA libraries provided in `pyplotting:latest-cuda11` container), and a GPU is present, then `pyhf` will use the `jax` backend and perform the calculations using the GPU. In the example below, the `-c` option tells `muscan.py` to not use a GPU even if it is available (and to use the CPU instead), while the `-b jax` option tells `muscan.py` to use the `jax` backend for calculations.

```
[pyhf]
script = "muscan.py"
skip = false
likelihood = "Slepton_bkgonly.json"
image = "pyplotting:latest"
gpu-options = "-c -B jax"
other-options = ""
```
Listing 20: The `pyhf` block of an example TOML configuration file for generating slepton events.