# Peer review of "Reduce, Reuse, Reinterpret: an end-to-end pipeline for recycling particle physics results"

_SciPost Physics Codebases, doi:SciPost Phys. Codebases 27-r0.5 (2024) , SciPost Phys. Codebases 27 (2024)_

## Round 1 · Referee Report · Anonymous (Referee 1) · 2023-10-9

Strengths

1-Clear and well written
2-Example use case clearly illustrate the package
3-Code and documentation included
4-Grammatical error free

Weaknesses

1-Not really suitable for scipost physics

Report

This manuscript introduces a Python package "mapyde", that combines existing toolkits into a workflow for 'reusing' past analysis on new models. Package usage is demonstrated through an SUSY example using previous ATLAST results.

The source code is included in a separate Github repository. The paper is very well-written and easy to follow. There is no grammatical errors. The example included clearly demonstrates the power of the proposed pipeline. I do not have any comment here.

Since the included analysis is not new, and the individual packages have been used extensively in the literature, I think this addition would be perfect for publication under SciPost Physics Codebases rather than SciPost Physics.

Requested changes

1-"... In either case, ... are only strictly .... models only ...": delete redundant "only"
2-"python" and "github" should be "Python" and "GitHub"

---

## Round 1 · Referee Report · Anonymous (Referee 2) · 2023-10-27

Report

The paper "Reduce, Reuse, Reinterpret: an End-to-End Pipeline for Recycling Particle Physics Results" by G. Stark, C. Ots and M. Hance presents the MaPyDe toolkit for reinterpretation of analyses based on the SimpleAnalysis framework for analysis description. The work is interesting and the tool is clearly a useful addition to the already existing reinterpretation tools. Physics applications of the toolkit to a few BSM scenarios are also briefly discussed.
The paper, however, does not present enough groundbreaking physics results.
In particular, similar scenarios have been investigated in other pMSSM studies and the single ATLAS analysis considered in the work is also available in other reinterpretation tools (such as CheckMATE and SModelS).
Therefore I strongly recommend it to be submitted to SciPost Physics Codebases with some minor modifications.
Below I list some of changes which I think would improve its readability and usefulness for MaPyDe users.

Requested changes

In order to make the draft more suitable for SciPost Physics Codebases, I recommend the following changes: 1. I suggest to move the information presented in Appendix B to the main body of the text, since it is an essential information for running the toolkit. 2. It would also be desirable to include a short section with installation instructions. Although this can be found it the project page, it could be included for completeness. 3. Finally, more details about the running options should be provided.

In addition, the following points should be addressed: a. The url for the repository containing the paper data (Ref.[42]) is wrong: "-atlas" is missing. b. In Figure 1: the SLHA input is shown as coming from the "Analysis Paper+HEPData" box. Why is it not part of the basic MadGraph input (which usually requires a process, parameter and run cards)? c. For completeness Figure 4 should include the value of slepton mass used as well as the value of $\Delta m(\tilde l,\tilde \chi_1^0)$. d. The link to the containers (https://github.com/scipp-atlas/container_registry) seems to be broken. e. Both in Figure 2 and 7 it would be desirable to show the ratio between the cross-section upper limits obtained from MaPyDe and the official ATLAS ones across the mass plane. This provides a more robust validation for a wide range of masses, since it goes beyond the small region of parameter space around the exclusion curve. f. "Listing A" is mentioned at the top of page 10, but it should probably read "Listing 1". g. It is very briefly mentioned that dark matter and the muon $g-2$ constraints are imposed. But more details should be provided. In particular, which range of $g-2$ values was considered? Also, which dark matter constraints were imposed?

---

## Round 1 · Referee Report · Anonymous (Referee 3) · 2023-10-30

Report

The authors describe in this article the so-called mapyde tool chain. This tool aims at facilitating the reinterpretation of existing analyses by the LHC collaborations. This looks like a rather interesting and easy to use tool which in principle warrants publication. However, there is one aspect which is not entirely clear: the authors present their tool for the use of one analysis only but what make simplified model analyses really interesting is that they can be combined in principle to investigated more realistic models. Thus, can one use this tool for the combination of different analyses? If yes, the authors should briefly discuss how their tool can be used to get constraints on a model from the combination of different analyses. This could be done in an additional appendix.

Moreover, there are a few more minor things that should be clarified. 1- page 4, 1st paragraph of section 3.2: It should be mentioned that the NLO corrections used by the ATLAS collaboration are QCD corrections but do not include electroweak corrections.

2- page 4, 2nd paragraph of section 3.2: the authors mention that they need to adjust the lepton efficiencies in DELPHES to reproduce the ATLAS signal yields. They state that the values used correspond to the upper range of their ref.[41]. It would be helpful if these numbers are collected in a short appendix so that the paper is self-contained.

3- page 4, 3rd paragraph of section 3.2: the authors state, that they consider a scenario in which a slepton decays into a wino-like neutralino assuming that the corresponding branching ratio is 100 per-cent. This assumption is unrealistic as one would also expect a wino-like chargino with similar mass as the neutralino in such case. This would also contribute to their signal albeit with different kinematics. While the assumption is fine for the purpose of the demonstration that the authors have mind, they should mention either in the text or as footnote that this assumption is not realistic as outlined in the comment here.

4- page 9, fig.8.: from the figure one could the impression that the authors intend to study decays with an on-shell W-boson while according to the discussion in the text it has to be an off-shell W-boson. I propose that the authors replace W-boson' in the caption by(off-shell) W-boson' to avoid any confusions.

5- figure 9 and the corresponding discussion: I do not fully understand what is shown here nor the caption. a) - the authors consider also scenarios where both charginos are presented as indicated in fig.(8). In this case there will be four neutralinos present and not only three as claimed in the caption. b) - What is the precise meaning of `fraction of models excluded'? What do we learn from this number given the fact that each bin contains only a few points and it is not clear how representative these points are. c) Why do the authors give only the line for the higgsino case and not also the one for the bino/wino case of their ref.~[41]? In their scan both cases are possible and, thus, both lines should be shown.

Requested changes

1- please add a discussion on how to combine different analyses to constrain a model

2 - please add the lepton efficiencies used for DELPHES, see comment 2 above

3- figure 9, please add the line for the exclusion of the wino-bino scenario from ref [41]

---

## Round 2 · Author Response

Dear colleagues,

Thank you for your helpful comments on our paper. We have considered all of them carefully, and provide responses to the individual reviewers below. In most cases we have updated the text or figures to either clarify or extend the results as suggested. We agree with the reviewers’ suggestion of submission to SciPost Physics Codebases, and are happy to be resubmitting our revised manuscript.

Referee #1:
Thank you for your helpful comments. We have implemented both suggested changes.

Referee #2:
Thank you for your useful suggestions. We implemented most of the recommended changes, including fixing all broken links and minor changes to figures.

We prefer not to move material from Appendix B into the main body of the text, since the tool described there is not part of the mapyde toolkit. However, we did add installation instructions, and some notes on running options and example commands.

Figure 1 was updated according to your comments, as was Figure 4. The label for Listing A/1 was corrected.

We added heatmaps to Figures 2 and 7, as suggested – this took some effort, and required a reprocessing of the mapyde Higgsino grid to align with the ATLAS grid, but we agree this provides a much more robust validation of the analysis chain. Thank you for this suggestion!

Finally, we clarified the implementation of DM/g-2 constraints in the text.

Referee #3:

Thank you for your useful comments. The first and most substantive point raised in the summary and list of requested changes concerns combinations of searches, which we agree is an interesting and important aspect of the LHC search program, particularly for analyses that use simplified models. However, it is not something that the mapyde toolkit was constructed to address. We elaborate more in our detailed responses below, and we will think carefully about whether and how a tool like mapyde can facilitate combinations of searches in the future, but that will have to be addressed in a future work.

We have added text to the paper clarifying the nature of the NLO corrections, and added an appendix with the delphes lepton efficiencies used in the reinterpretation.

We agree with the reviewer that the slepton-wino-bino grid is not necessarily realistic, though we still feel the reinterpretation of the ATLAS searches in this model is useful. We agree that realistic models would likely include slepton decays through charginos. We ignored them because the branching fractions necessary to produce 2 opposite-sign, same-flavor leptons are far less favorable in the chargino decay channel, and the corresponding leptons are significantly softer, leading to worse reconstruction efficiencies and overall very small event yields in the ATLAS signal regions. We also wanted to keep the differences with respect to the ATLAS simplified model as small as possible, and in the case where the N2-N1 splitting vanishes, the slepton-wino-bino model considered here converges to the slepton-bino model from the ATLAS note. We have added a footnote with this information.

We have modified Figure 8 to clarify the final state and avoid implying that the W bosons mediating the chargino decays are on-shell.

We have also clarified the caption of Figure 9 to reflect the correct number of light neutralinos, and added the ATLAS WinoBino exclusion contour to the plot. We agree that the “fraction of models excluded” may not be a perfect metric for the strength of the analysis, but it is a common way to demonstrate the reach of a search when scanning over pMSSM model parameters.

Thank you all once again.
With best wishes,
Giordon, Camila, and Mike

---

## Round 2 · List of Changes

• Implemented both suggested text changes from Reviewer #1
  • Added instructions for installing the python package, and example command-line calls to the toolkit
  • Fixed broken URL’s
  • Updated Figure 1 to include the param card as part of the user configuration
  • Added text to figure 4 to clarify masses
  • Remade figure 2 and 7 to include heatmaps reflecting the difference between the ATLAS and mapyde results across the full grid, not just near the contours. This required reproducing the Higgsino grid to match the ATLAS grid spacing, which resulted in some minor changes to the mapyde exclusion contour.
  • Fixed labeling for Listing A, and re-added a previously omitted Listing.
  • Clarified text on dark matter constraints
  • Clarified nature of NLO corrections used by ATLAS as suggested by Reviewer #3
  • Added an appendix with delphes efficiency maps used in the reinterpretation
  • Added a footnote explaining the choice of a 100% branching ratio of sleptons decaying to N2+lepton.
  • Updated figure 8 to replace W bosons in final state with their decay products
  • Fixed caption in Figure 9 to say “at least three light” neutralinos
  • Added ATLAS WinoBino exclusion contour to Figure 9
  • Additional minor revisions to the text for accuracy and clarity.

---

## Editorial Decision

published